# *Ganoderma lucidum* Modulates Inflammatory Responses following 1-Methyl-4-Phenyl-1,2,3,6-Tetrahydropyridine (MPTP) Administration in Mice

**DOI:** 10.3390/nu14183872

**Published:** 2022-09-19

**Authors:** Zhili Ren, Hui Ding, Ming Zhou, Piu Chan

**Affiliations:** 1National Clinical Research Center for Geriatric Disorders, Xuanwu Hospital of Capital Medical University, Beijing 100053, China; 2Department of Neurobiology, Neurology and Geriatrics, Xuanwu Hospital of Capital Medical University, Beijing 100053, China; 3Advanced Innovation Center for Human Brain Protection, Capital Medical University, Beijing 100053, China; 4Clinical Center for Parkinson’s Disease, Capital Medical University, Key Laboratory for Neurodegenerative Disease of the Ministry of Education, Beijing Key Laboratory for Parkinson’s Disease, Beijing 100053, China; 5Beijing Institute of Brain Disorders, Collaborative Innovation Center for Brain Disorders, Capital Medical University, Beijing 100053, China

**Keywords:** *Ganoderma lucidum*, Parkinson’s disease, neuroinflammation, microglial activation, microglial autophagy

## Abstract

*Ganoderma lucidum*, one of the most valued medicinal mushrooms, has been used for health supplements and medicine in China. Our previous studies have proved that *Ganoderma lucidum* extract (GLE) could inhibit activation of microglia and protect dopaminergic neurons in vitro. In the present study, we investigated the anti-neuroinflammatory potential of GLE in vivo on Parkinsonian-like pathological dysfunction. Male C57BL/6J mice were subjected to acute 1-methyl-4-phenyl-1,2,3,6-tetrahydropyridine (MPTP) lesion, and a treatment group was administered intragastrically with GLE at a dose of 400 mg/kg. Immunohistochemistry staining showed that GLE efficiently repressed MPTP-induced microglia activation in nigrostriatal region. Accordingly, Bio-plex multiple cytokine assay indicated that GLE treatment modulates abnormal cytokine expression levels. In microglia BV-2 cells incubated with LPS, increased expression of iNOS and NLRP3 were effectively inhibited by 800 μg/mL GLE. Furthermore, GLE treatment decreased the expression of LC3II/I, and further enhanced the expression of P62. These results indicated that the neuroprotection of GLE in an experimental model of PD was partially related to inhibition of microglia activation in vivo and vitro, possibly through downregulating the iNOS/NLRP3 pathway, inhibiting abnormal microglial autophagy and lysosomal degradation, which provides new evidence for *Ganoderma lucidum* in PD treatment.

## 1. Introduction

Parkinson’s disease, the second most common neurodegenerative disorder, is mainly characterized by motor dysfunctions and non-motor symptoms. Disturbances caused by PD aggravate patients’ physical and mental state. Although enormous treatment studies have been conducted in PD, current therapeutic strategies can only manage the symptoms but neither modulate nor modify pathological alteration or neurodegenerative process. It is well noted that many agents with great promise in the laboratory have not been validated in clinical studies [1,2]. Thus, exploring candidates and druggable compounds capable of delaying or halting progression of PD is constantly and greatly needed. Multiple lines of evidence show that neuroinflammation induced by microglia activation and/or mitochondrial dysfunction is closely associated with disease progression, thus making it a potential therapeutic target for treating PD.

Traditional Chinese medicine has been extensively and thoroughly studied in the management of neurological diseases. *Ganoderma lucidum* (GL), a well-known culinary and medicinal mushroom which is widely used in Asia (especially in China), is beneficial for health care and health longevity [3]. They are rich in nutritional, bioactive, and pharmacological active compounds such as polysaccharides, triterpenoids, nucleosides, sterols, alkaloids, amino acids, and a variety of microelements. Thus far, accumulating studies mainly focus on observing bioactive compounds of *Ganoderma lucidum* and multi-protective effects against neurological disorders. As reviewed by Phan, as a typical representative of fungi, GL has shown neuroprotective properties such as promoting neuronal survival and neuritogenesis both in vitro and in vivo, indicating underlying improvement in neurological recovery and function [4].

Many studies on anti-inflammation of *Ganoderma lucidum* have been thoroughly researched and discussed. *Ganoderma lucidum* polysaccharides (GLPs), one of the most valuable bioactive compounds in GL, has been reported in the literature to modulate inflammation against various diseases. GLPs extracted from the sporoderm-removed spores of GL alleviated azoxymethane/dextran sulfate sodium-induced colitis and tumorigenesis in mice, and inhibited lipopolysaccharides (LPS)-induced inflammation markers and MAPK activation in vitro [5]. A recent study reported that GL product containing a mixture of spores and fruiting bodies, namely, “GLSF”, had anticancer activity in vivo through attenuating inflammation, NF-kB, and/or KRAS activation [6]. Another bioactive compound, *Ganoderma lucidum* triterpenoids (GLTs), can also exert an anti-inflammation effect. Chen et al. newly found GLT administration could mitigate maternal separation-induced anxiety- and depression-like performance via reducing peripheral and cerebral inflammation, with non-toxic effect on the key organs [7]. The safety of *Ganoderma lucidum* has also been paid more attention. Biomass extract and exopolysaccharide of GL were considered as non-toxic for vertebrates [8,9].

Our team previously revealed that *Ganoderma lucidum* extract (GLE) could suppress microglia-derived proinflammatory generated by Lipopolysaccharide (LPS) and 1-methyl-4-phenylpyridinium (MPP+) in cultures of microglia or MES23.5 cells [10,11]. Furthermore, we employed MPTP-lesioned acute mouse model of PD and found that GLE treatment could significantly improve motor ability and dopaminergic function in nigrostriatal pathway [12,13]. However, it remains unknown whether GLE mitigates neurodegenerative pathologies by modulating neuroinflammation. Although numerous studies have reported the molecular mechanisms of GL, there is still a lack of clear molecular mechanisms of anti-inflammation, especially in PD models. In the present study, we aim to investigate the neuroinflammation effect of *Ganoderma lucidum* in Parkinsonian model mice, providing beneficial data for its neuroprotective and neuroinflammatory regulation.

## 2. Materials and Methods

### 2.1. Reagents

GLE was kindly provided by PuraPharm Corporation (Guangxi, China). In detail, GL fruiting bodies were collected from Guizhou province. GLE was produced as outlined in Figure 1. First, the dried fruiting bodies of GL were broken into small pieces. Then, a certain amount of fruiting bodies was extracted in the conditions as follows: 80% ethanol twice, extraction time of 2 h, extraction temperature of 80 °C, and the residues were then extracted with 50% ethanol twice. The residues were then followed by 2 h boiling water extraction procedure for two repeated times. The above steps were carried out in the traditional Chinese medicine extraction tank, which can automatically adjust the temperature. Finally, the supernatant was combined, centrifuged, and lyophilized to dryness. GLE was standardized to contain about 9.8% *w*/*w* polysaccharides, 0.3–0.4% *w*/*w* Ganoderic acid A, and 0.3–0.4% Ergosterol. GLE was prepared freshly using corresponding vehicle (suspended in 0.5% CMC-Na in vivo or dissolved in cultured medium in vitro) in the dark.

### 2.2. Animals

Adult (8-week-old) male C57BL6/J mice were purchased from Beijing Vital River Laboratory Animal Technology (Beijing, China). Animals were maintained under standard conditions of temperature and humidity with a 12 h light/dark cycle and fed a standard pellet diet and water ad libitum. All animals were treated in strict accordance with the NIH Guide for Care and Use of Laboratory Animals and the study was approved by Animal Care Committee of Capital Medical University, approval Code: XW-20210425-2, approval Date: 25 June 2021.

### 2.3. Experimental Procedure

After acclimatization for one week, 45 male mice were randomly divided into three groups (*n* = 15) of Control, MPTP(Sigma-Aldrich(Shanghai) Trading Co., Ltd, Shanghai, China), MPTP + GLE. Briefly, an acute model of PD was conducted with MPTP (20 mg/kg) intraperitoneal injection to the mice four times at 2 h intervals in 1 day. The next day, mice in both Control and MPTP groups received equal amounts of 0.5% CMC-Na solution once daily for 4 weeks successively. Animals in the MPTP + GLE group were treated with intragastric administration of 400 mg/kg GLE.

### 2.4. Immunohistochemistry (IHC) and Imaging Analysis

After 4 consecutive weeks of administration, mice were transaortically perfused, firstly with 0.1 M PBS, then 4% paraformaldehyde (pH 7.4). Brains were carefully removed from the skull and placed within 4% formaldehyde in PBS overnight, then cryoprotected in 15% sucrose (*w*/*v* PBS), and then, 20% and 30% sucrose. Sections were stored free-floating at −20 °C in cryoprotectant (25% *v*/*v* glycerin, 25% *v*/*v* ethylene glycol; 50% *v*/*v* 0.05 M phosphate buffer) until staining.

In brief, coronal sections (35 μm) were collected and heat- or trypsin-mediated antigen retrieval was performed before commencing with IHC staining protocol. Then, sections were permeabilized in 0.3% PBST for 30 min and quenched endogenous peroxidases using hydrogen peroxide. Brain sections were blocked in 10% fetal bovine serum for 1 h and then incubated overnight at 4 °C, with rabbit anti-ionizing calcium-binding adaptor molecule 1 (Iba1), (1:1000, Wako, specific for microglia). Finally, the slices were mounted and captured using a CCD camera (Olympus, Tokyo, Japan)). Images were analyzed using Image J software.

### 2.5. Multiple Assays of 23 Cytokines in Brain Tissues

Cytokine detecting was performed using Bio-Plex Cytokine Assay Kit (Bio-Rad Laboratories, Hercules, CA, USA). Briefly, midbrain and striatum of mice in each group were rapidly removed for the preparation of brain supernatant homogenates. The remaining steps were carried out strictly according to the manufacturer’s instructions. We looked at a panel of 23 cytokines and chemokines, including IL-1α, IL-1β, IL-2, IL-3, IL-4, IL-5, IL-6, IL-9, IL-10, IL-12(p40), IL-12(p70), IL-13, IL-17, Eotaxin, G-CSF, GM-CSF, KC, MCP-1, MIP-1α, MIP-1β, RANTES, TNF-α, and IFN-γ.

### 2.6. Cell Culture

The microglia cell line BV-2 was maintained at 37 °C in Dulbecco’s Modified Eagle’s Medium (DMEM) supplemented with 10% fetal bovine serum and 100 U/mL penicillin and streptomycin in a 5% CO_2_ humidified incubator.

### 2.7. Cell Viability Assay

The BV-2 proliferative activity was assayed using CCK-8 kit (Dojindo Molecular Technologies, Kyushu, Japan). Briefly, cells were seeded in 96-well culture plates and incubated with different concentrations of GLE in the presence of LPS (1 μg/mL) for the indicated time. At the end point, the culture medium was removed and the cells were washed with PBS, and then incubated with CCK-8 reagent for an additional 2 h. CCK-8 was diluted with cultured medium to form a 10% CCK-8 solution. The absorbance was measured at 450 nm wavelength. Cell viability was evaluated as the ratio of the sample to that of control.

### 2.8. Western Blot Analysis

The cellular samples were lysed in RIPA lysis buffer containing a protease and phosphatase inhibitor cocktail (Applygen Technologies Inc., Beijing, China) and centrifuged at 12,000× *g* for 15 min at 4 °C. The protein concentration of cell supernatant was determined with BCA method. Briefly, equal amounts of proteins were loaded and transferred onto the PVDF membrane. Then, membranes were incubated in blocking solution with 5% milk at RT for 1 h and then incubated at 4 °C overnight, with the following primary antibodies: inducible nitric oxide synthase (iNOS), NOD-like receptor thermal protein domain-associated protein 3 (NLRP3), LC3B, and SQSTM1/P62 (1:1000, Cell Signaling Technology, Danvers, MA, USA) in TBST containing 5% milk. The membranes were incubated with the corresponding secondary antibodies for one hour and washed with TBST for 30 min. Then, the membranes were imaged with the Chemiluminescence Imaging System (Tanon 5200, Shanghai, China). β-actin was used as a loading control in all Western blot analyses. The quantification of blot intensity was performed using Image J software.

### 2.9. Statistical Analysis

Data were statistically analyzed using one-way analysis of variance followed by LSD or Dunnett’s T3 post hoc test, depending on the homogeneity of the variance test. Summarized data were expressed as mean ± SEM. Statistical significance was set at *p* < 0.05. All statistical analysis was undertaken using SPSS v17.

## 3. Results

### 3.1. GLE Treatment Attenuated Microglia Activation in MPTP-Lesioned Mice

IHC assay was used to evaluate the effects of GLE on the microglia activation in lesioned regions of MPTP-lesioned mice. We used design-based stereology to count the number of Iba1-positive cells. Notably, we found that, compared with the control group, the number of Iba1-positive cells was increased in the striatum regions of the MPTP model group. However, GLE-treated animals evidenced a marked decrease in the number of Iba1+ cells, as shown in Figure 2A,B. Similarly, as shown in Figure 3A,B of the substantia nigra (SN) region, our results showed that Iba1 was predominantly expressed by the microglia and Iba1 levels were significantly higher in the MPTP group, while GLE treatment significantly inhibited the expression of Iba1 compared with the MPTP group. The changes in SNpc among groups coincided well with those in SNpr, but the number of Iba1+ cells in SNpr is about three times more than that of the SNpc region.

### 3.2. GLE Treatment Modulated Cytokine and Chemokine Levels in MPTP-Lesioned Mice

In order to provide additional evidence of the effect of GLE on inflammation following MPTP neurotoxicity, we performed bioplex assay to quantitate the expression levels of inflammatory mediators. Table 1 and Table 2 showed the cytokine concentrations in the brain-damaged regions after MPTP neurotoxicity with or without GLE treatment. Our results (Table 1) indicate that the levels of cytokines (IL-1β, IL-12(p40), IL-17, GM-CSF, KC, MIP-1β, and TNF-α) and interferon (IFN-γ) were up-regulated in the mesencephalon of MPTP-injured mice compared to that of control mice. GLE treatment down-regulated GM-CSF levels compared with MPTP group. No significant changes were found in the levels of IL-1β, IL-12(p40), IL-17, KC, MIP-1β, TNF-α, or IFN-γ in response to GLE treatment. Unlike the midbrain, most of the cytokines (especially some important proinflammatory cytokines such as IL-1α, IL-1β, IL-2, IL-4, IL-12, TNF-α, and IFN-γ) had high expression levels in the striatum of MPTP mice compared with levels in the striatum of control mice. Interestingly, whereas treatment with GLE led to declines in proinflammatory cytokine expression in the striatum, it significantly decreased the expression of IL-1α, IL-1β, IL-3, and TNF-α. Specifically, MPTP treatment also significantly augmented the expression of G-CSF, GM-CSF, and chemokines (KC, MIP-1α, and MIP-1β) in striatum regions (Table 2).

### 3.3. GLE Inhibited LPS-Induced BV2 Microglia Proliferation

We used an in vitro model of microglia proliferation, established by examining BV2 microglia proliferation in response to LPS stimulation at 6, 12, and 24 h. The proliferation rate of BV2 microglia appeared unaffected by LPS at 6 and 12 h. At 24 h, 1 μg/mL LPS-stimulated microglia displayed an approximate 50% increase in proliferation. Treatment with GLE decreased LPS-induced proliferation of BV2 microglia dose-dependently, and 800 μg/mL GLE induced a significant decrease compared with the LPS-stimulation-only group (Figure 4).

### 3.4. GLE Suppressed LPS-Stimulated iNOS-NLRP3 Activation in BV-2 Cells

We sought to determine the role of iNOS-NLRP3 in GLE-mediated immunomodulation of microglial responses. We further evaluated iNOS and NLRP3 expression in LPS-stimulated BV-2 microglia. In normal condition culture, BV2 microglia produced negligible amounts of iNOS and NLRP3. LPS stimulation significantly increased the expressions of both proteins at 24 h. Conversely, the presence of GLE reduced iNOS level compared with LPS-stimulated BV2 cells alone, similar to the expression level of unstimulated BV2 microglia. In the same way, treatment with 800 μg/mL GLE for 24 h significantly attenuated LPS-induced upregulation of NLRP3 expression (Figure 5).

### 3.5. GLE Inhibited Autophagy and Lysosomal Degradation in LPS-Induced BV-2 Cells

Autophagy-related proteins LC3 and P62 are often used to evaluate autophagic level in vivo and vitro. Therefore, we detected expression levels of these two markers. Our results showed that, compared with the control group, the expression of LC3II/I in the LPS group increased, but has no statistically significant difference. P62 expression of the LPS group also increased, suggesting the lysosome could not bind to the autophagosome at this time. However, in the LPS + GLE group, P62 further increased, but LC3II/I decreased compared with the LPS group (Figure 6).

## 4. Discussion

To our best knowledge, this is the first study to observe the anti-neuroinflammatory effect of *Ganoderma lucidum* in Parkinsonian pathogenesis in vivo. Our findings showed that GLE could inhibit microglia activation and modulate cytokine levels in MPTP-neurotoxicity parkinsonian mice. In addition, our in vitro study revealed that GLE could inhibit microglial activation by suppression of iNOS/NLRP3 activation and mitigate dysfunction of autophagy and lysosomal degradation induced by LPS in BV-2 microglial cells.

Mitochondrial neurotoxin MPTP contributes to the neurodegenerative loss of dopaminergic neurons in the substantia nigra pars compacta and striatum. Its neurotoxicity is generally attributed to a cascade of deleterious events, such as mitochondrial dysfunction, apoptosis, inflammation, ubiquitin-proteasomal system (UPS) dysfunction, and oxidative stress, ultimately leading to neuronal damage [14,15]. Increasing evidence indicates the involvement of microglia in MPTP neurotoxicity [16]. Histology and counting results showed that MPTP causes dramatic activation of microglia in striatum and SN, which is consistent with previous studies [17,18]. Herein, we captured and counted both SNpc and SNpr sections. We found the microglia density of SNpr is three times that of SNpc, which may suggest that axons and dendrites in SNpr are vulnerable to inflammatory damage compared to somas in SNpc. We speculate that regional distinct distributions of microglia largely depend on physical and pathological conditions, being close to and interacting with highly neural activity. This was also proved by exhibiting basal differences in cytokine profiles and differential cytokine upregulation with MPP+ in regionally isolated microglia (SN, ventral tegmental area, and cortex) [19]. Furthermore, microglia actively participate in regulating neuronal excitability and function. Early axon loss also renders neurons dysfunctional [20]. Our findings verify that GLE with 9.8% polysaccharide treatment obviously inhibit microglia activation in both striatum and SN sections, confirming its potential to regulate neuroinflammation of PD.

Production and release of cytokines plays a central role in the microglia-mediated inflammatory action [21]. Abnormally reactive microglia and dysregulated cytokine levels are often concomitant with alterations in synaptic structure and function [22]. Our in vivo study indicated that GLE modulates abnormal cytokine levels to different extents in striatum. For example, altered levels of cytokines such as IL-1α, IL-1β, MIP-1α, and TNF-α have been corrected to nearly normal levels following MPTP challenge. Another extract from GL (GLE Cat# 1288372) pre-treatment inhibits the expression of the pro-inflammatory cytokines G-CSF, IL1a, MCP-5, MIP3α, and RANTES in the LPS-stimulated BV-2 cells [23]. Thus, its effect is beneficial for abnormally reactive microglia, especially for the synaptic functional output of substantia nigra striatum pathway, exhibiting enhancement of motor ability [12]. Considered together, immunohistochemistry staining combined with the Bioplex data suggest that GLE could modulate abnormal microglial activation and cytokine expression levels caused by MPTP, indicating its improvement of mitochondrial function and microglia function.

Next, we conducted CCK-8 assay to test GLE on proliferative activity of BV-2 treated with LPS. Contradictory to the viability data obtained by Hilliard A et al. using Alamar Blue^®^ (Resazurin) assay [23], LPS significantly promoted proliferative activities, while GLE 800 ug/mL reduced viability in our BV-2 cellular model. In addition, many studies have shown that LPS (1 μg/mL) absolutely does not affect the proliferation activity of BV-2 cells [24,25].

iNOS, as one of the key proinflammatory biomarkers, could hardly be detected in mRNA and protein expression at baseline. However, it is induced and up-regulated under inflammation, such as LPS-treated animal or microglial cells, contributing to neurodegeneration [26]. As predicted, GLE completely blocked LPS-induced overexpression of iNOS and NLRP3, as determined by Western blotting. The NLRP3 inflammasome has been studied extensively and was found to be activated by a series of stimuli [27]. The NLRP3 inflammasome and its role in PD progression have been attractive subjects recently. Han X and colleagues reported that NLRP3 inflammasome activation correlates with PD progression, and could be inhibited by kaempferol via ubiquitination and autophagy, indicating that targeting NLRP3 is a promising therapeutic strategy for PD [28]. Whether GLE directly or indirectly targets NLRP3 needs to be explored in further studies.

The mechanism via which GLE can regulate neuroinflammation remains unclear. The role of microglial autophagy in microglial function has gained increasing attention [29]. Microglia played a neuroprotective role via selective autophagy in the clearance of neuronal α-synuclein [30]. Microglia-specific deletion of Atg7 enhances intraneuronal tau pathology and its spreading, revealing an essential role for microglial autophagy in regulating neuroinflammation [31]. LC3 is required for the formation of autophagosome membranes and is transformed from LC3-I to LC3-II during autophagy activation, the latter of which is considered a marker of autophagosomes in mammalian cells. Sequestosome-1 (SQSTM1)/P62 is a poly-ubiquitin-binding protein that is degraded via autophagy. Herein, we detected the expressions of LC3 and P62, which are major components involved in the autophagy process [32]. LPS increased LC3-I transformation to LC3-II and increased P62 expression, resulting in abnormally high autophagy flux. Interestingly, GLE could decrease LC3-I transformation to LC3-II, and meanwhile inhibit P62 degradation, indicating its potential for balancing between autophagy formation and degradation [33]. These data provide a new insight of GLE in neuroprotection by inhibiting NLRP3 inflammasome and abnormal autophagic processes in microglial cells.

Which ingredient(s) of GL that perform its function in neuroinflammatory modulation is also unclear. As we know, GL is highly rich in hundreds of bioactive components. Any component may exert anti-neuroinflammatory effects. *Ganoderma lucidum* polysaccharides (GLPs), one of the effective fractions, has been proven to play regulatory roles in LPS- and Aβ-mediated neuroinflammation in an in vitro study, demonstrating a potent modulator for AD-related neuroinflammation [34]. Accumulating in vivo studies demonstrated that GLPs could also exert neuroinflammatory modulation in other various neurological disorders such as in D-galactose rats and spinal cord ischemia–reperfusion injury rats [35,36]. Ganoderic acid A (GAA), another popular extract, could inhibit LPS-induced neuroinflammation in BV2 microglial cells and modulated neuroimmune in two multiple sclerosis models via activating FXR receptor [37,38]. It was suggested that GAA could also suppress inflammation via regulating M1/M2 microglial polarization in a rat model of post-stroke depression [39]. Other components of GL were further extracted and confirmed anti-neuroinflammation action in the classical LPS-stimulated BV2 cell model [40,41]. Generally speaking, different constituents are largely determined by different extraction methods. Thus, different components may exert similar pharmacological effects through different pathways. The aqueous extract of *Ganoderma lucidum* was found to decrease immunoreactivity for GFAP as well as TNF-alpha and IL-1beta in the CA3 region in rats induced by kainic acid [42]. In a word, there is no consensus on which components mediate this pathological process thus far. We prefer that some components play a synergistic role in the body, which we mentioned 10 years ago [43]. We have surveyed the recent studies of *Ganoderma lucidum*, and provided a comparison table of available anti-neuroinflammatory extracts, beside Parkinsonian pathogenesis (Table 3), which be better for us to understand the anti-neuroinflammatory components of *Ganoderma lucidum*.

## 5. Conclusions

In summary, our results reveal that GLE prevents MPTP-induced neuroinflammation, and an in vitro study thus provides the implication that anti-inflammatory mechanism may be partially involved in suppression of NLRP3 activation and microglial autophagy deficiency, suggesting that *Ganoderma lucidum* could be chosen for targeting microglia as a therapeutic intervention in PD.

## Figures and Tables

**Figure 1 nutrients-14-03872-f001:**
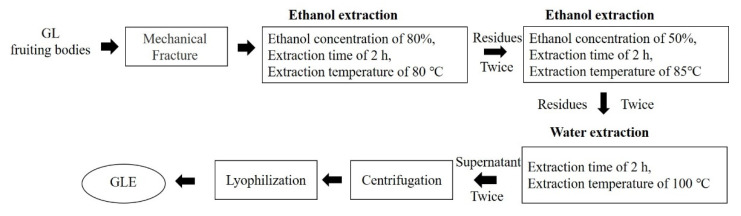
GLE preparation scheme.

**Figure 2 nutrients-14-03872-f002:**
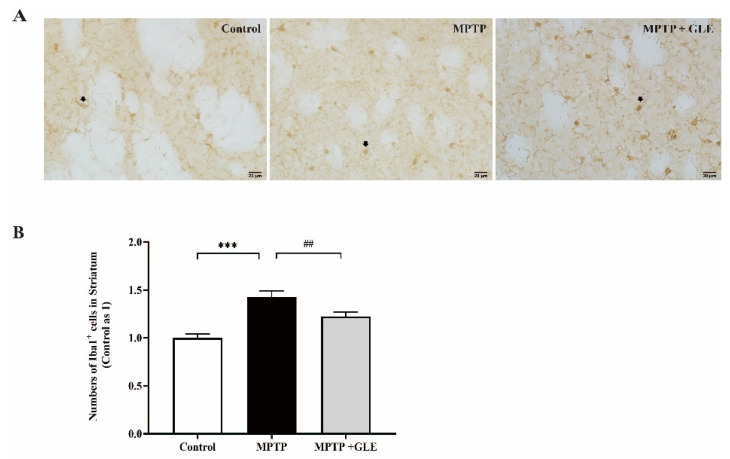
Effect of GLE on the expression of Iba1 in striatum of PD mice. (**A**) Representative immunohistochemistry images of Iba1 immunostaining in striatum, scale bar = 20 um. (**B**) The statistical results of Iba1-positive cells in the striatum area of each group. ***: *p* < 0.001, Control group vs. MPTP group; ##: *p* < 0.01, MPTP group vs. MPTP + GLE group. Data are presented as mean ± SEM. Each group *n*/N ≥ 20 images/5 mice. The arrows stand for Iba1+ cells.

**Figure 3 nutrients-14-03872-f003:**
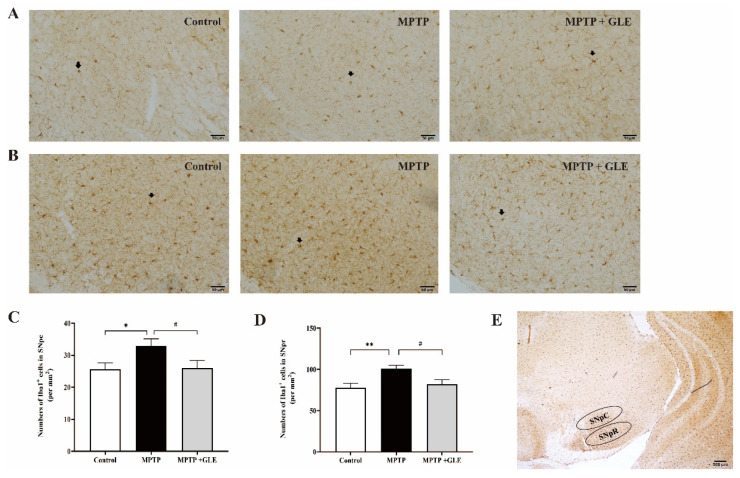
Effect of GLE on the expression of Iba1 in substantia nigra of PD mice. (**A**) and (**B**) Representative immunohistochemistry images of Iba1 immunostaining in substantia nigra pars compacta (SNpc) and substantia nigra pars reticulata (SNpr), scale bar = 20 um. (**C**) and (**D**) The statistical results of Iba1-positive cells in the SNpc and SNpr of each group, respectively. */**: *p* < 0.05/0.01, Control group vs. MPTP group; #: *p* < 0.05, MPTP group vs. MPTP + GLE group. Data are presented as mean ± SEM. Each group *n*/N ≥ 20 images/5 mice. The arrows stand for Iba1+ cells. (**E**) Diagram of subregion in substantia nigra, scale bar = 200 um.

**Figure 4 nutrients-14-03872-f004:**
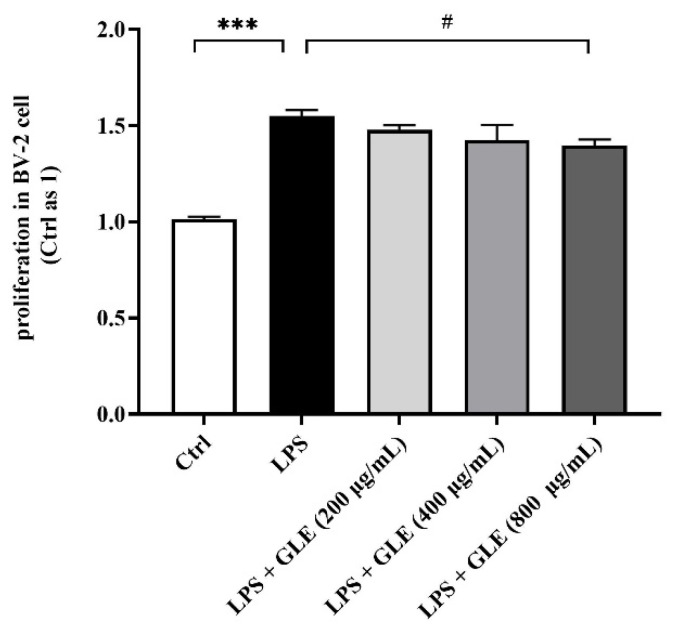
The effect of GLE on proliferation activity in LPS-induced BV-2 cells. Values are expressed as mean ± SEM of BV2 proliferation activity from three independent experiments. ***: *p* < 0.001, Ctrl group vs. LPS group; #: *p* < 0.05, LPS group vs. GLE treatment groups. All experiments were repeated at least in triplicate.

**Figure 5 nutrients-14-03872-f005:**
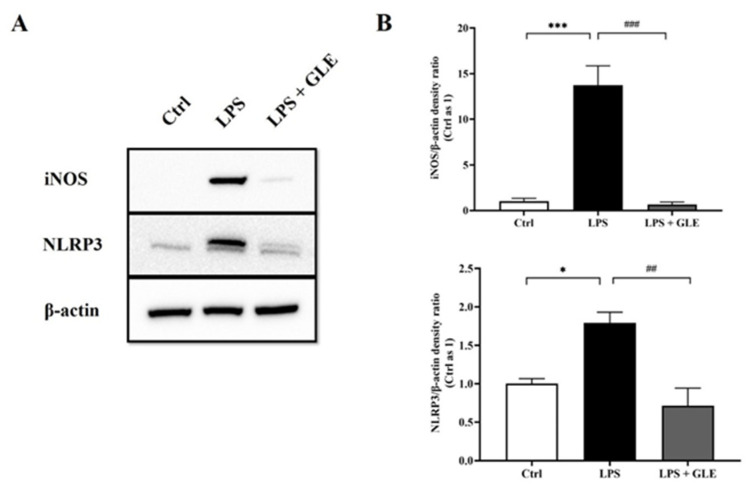
The effect of GLE on iNOS/NLRP3 expression in LPS-induced BV-2 cells. (**A**) Representative images of Western blot with antibodies against iNOS and NLRP3. (**B**) Quantification analysis of iNOS and NLRP3 expression from Western blot. Values are represented in the form of mean ± SEM. */***: *p* < 0.05/0.001, Ctrl group vs. LPS group; ##/###: *p* < 0.01/0.001, LPS group vs. LPS + GLE group. All experiments were repeated at least in triplicate.

**Figure 6 nutrients-14-03872-f006:**
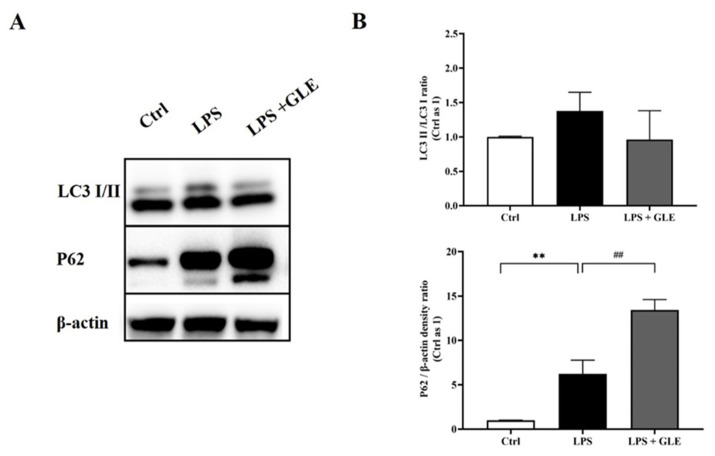
The effect of GLE on LC3 and P62 expression in LPS-induced BV-2 cells. (**A**) Representative images of Western blot with antibodies against LC3B and P62. (**B**) Quantification analysis ofWestern blot. The immunoblotting of LC3B was analyzed by examining the LC3-I-to-LC3-II conversion, expressed as the LC3-II/LC3-I ratio. Values are represented in the form of mean ± SEM. **: *p* < 0.01, Ctrl group vs. LPS group; ##: *p* < 0.01, LPS group vs. LPS + GLE group. All experiments were repeated at least in triplicate.

**Table 1 nutrients-14-03872-t001:** Mouse midbrain cytokine/chemokine concentrations ^1^.

Cytokine/Chemokine	Control	MPTP	MPTP + GLE
IL-1α	59.31 ± 7.63	69.06 ± 8.72	67.96 ± 4.83
IL-1β	5985.88 ± 513.24	7048.02 ± 329.99 *	6534.95 ± 176.21
IL-2	2006.17 ± 393.29	2437.60 ± 332.68	2421.81 ± 169.47
IL-3	162.65 ± 11.88	183.67 ± 8.04	169.14 ± 7.93
IL-4	53.63 ± 6.27	65.46 ± 8.04	63.02 ± 3.38
IL-5	123.02 ± 14.17	133.25 ± 9.50	135.11 ± 5.32
IL-6	174.53 ± 15.20	209.65 ± 7.87	198.62 ± 7.90
IL-9	26,312.90 ± 4622.58	30,082.75 ± 3523.41	28,480.24 ± 2334.98
IL-10	254.47 ± 26.70	280.79 ± 26.42	271.50 ± 16.68
IL-12(p40)	115.18 ± 8.51	135.87 ± 7.89 *	127.55 ± 3.78
IL-12(p70)	781.60 ± 84.75	794.76 ± 39.17	847.94 ± 30.29
IL-13	30,195.89 ± 2496.90	34,524.49 ± 1708.82	34,031.25 ± 871.92
IL-17	2384.74 ± 199.21	2820.02 ± 122.31 *	2737.16 ± 48.81
Eotaxin	14,680.31 ± 757.97	14,612.08 ± 289.31	13,475.91 ± 330.17
G-CSF	102.73 ± 11.11	116.59 ± 10.21	114.11 ± 5.11
GM-CSF	1555.74 ± 36.43	1651.85 ± 24.38 *	1514.10 ± 24.59 ##
KC	295.41 ± 16.34	367.05 ± 14.78 **	355.82 ± 15.63
MCP-1	1952.11 ± 128.24	2134.07 ± 128.52	2165.10 ± 86.53
MIP-1α	447.26 ± 39.74	441.25 ± 26.90	433.75 ± 21.50
MIP-1β	755.20 ± 89.44	936.23 ± 57.13 *	904.77 ± 42.74
RANTES	319.19 ± 33.62	341.21 ± 11.75	332.40 ± 6.73
TNF-α	5008.07 ± 156.57	6403.28 ± 334.58 *	6319.66 ± 253.61
IFN-γ	398.57 ± 38.14	481.55 ± 24.00 *	439.06 ± 26.87

^1^ All values are means ± SE (in pg/mL); *n* = 5–6 mice/group. */**: *p* < 0.05/0.01, Control group vs. MPTP group; ##: *p* < 0.01, MPTP + GLE group vs. MPTP group. IL, interleukin; IL-12(p40), IL-12 subunit p40; G-CSF, granulocyte colony-stimulating factor; GM-CSF, granulocyte-macrophage colony-stimulating factor; IFN, interferon; KC, keratinocyte chemoattractant; MCP-1, monocyte chemotactic protein-1; MIP, macrophage inflammatory protein.

**Table 2 nutrients-14-03872-t002:** Mouse striatum cytokine/chemokine concentrations ^1^.

Cytokine/Chemokine	Control	MPTP	MPTP + GLE
IL-1α	61.11 ± 3.82	75.43 ± 2.94 **	65.93 ± 2.18 #
IL-1β	4895.35 ± 459.92	5895.72 ± 161.64 *	4927.78 ± 138.20 #
IL-2	894.40 ± 69.15	1486.18 ± 218.39 *	1328.81 ± 69.23
IL-3	115.38 ± 8.88	138.77 ± 8.18	109.21 ± 8.20 #
IL-4	41.76 ± 3.42	54.33 ± 2.97 *	55.00 ± 4.52
IL-5	102.39 ± 10.97	120.48 ± 9.47	129.13 ± 9.16
IL-6	157.80 ± 18.02	183.73 ± 6.73	180.29 ± 7.09
IL-9	15,997.54 ± 1410.37	18,889.61 ± 1096.69	19,500.01 ± 878.82
IL-10	197.01 ± 16.39	244.15 ± 12.87 *	233.61 ± 16.56
IL-12(p40)	94.06 ± 7.31	115.49 ± 4.13 *	107.81 ± 5.11
IL-12(p70)	446.91 ± 34.71	531.65 ± 26.58 *	468.48 ± 18.29
IL-13	23,491.19 ± 2188.84	26,769.18 ± 1134.10	26,409.85 ± 939.14
IL-17	1787.37 ± 151.17	1921.06 ± 84.15	1990.46 ± 59.05
Eotaxin	7062.60 ± 160.99	7494.43 ± 295.88	7187.90 ± 159.94
G-CSF	80.60 ± 9.07	105.89 ± 7.23 *	100.30 ± 4.43
GM-CSF	725.62 ± 14.80	788.56 ± 14.17 **	771.08 ± 11.42
KC	250.23 ± 24.36	304.47 ± 14.22 *	306.95 ± 13.88
MCP-1	1448.53 ± 112.57	1592.19 ± 64.50	1660.90 ± 65.06
MIP-1α	262.39 ± 19.93	341.37 ± 17.93 **	311.37 ± 13.78 #
MIP-1β	361.47 ± 25.75	428.31 ± 18.00 *	388.70 ± 10.01
RANTES	227.98 ± 22.43	261.88 ± 11.39	242.62 ± 11.30
TNF-α	4535.07 ± 360.77	5566.64 ± 313.77 *	4606.82 ± 209.94 #
IFN-γ	339.27 ± 30.05	431.28 ± 20.21 *	390.81 ± 18.97

^1^ All values are means ± SE (in pg/mL); *n* = 5–6 mice/group. */**: *p* < 0.05/0.01, Control group vs. MPTP group; # *p* < 0.05, MPTP + GLE group vs. MPTP group. IL, interleukin; IL-12(p40), IL-12 subunit p40; G-CSF, granulocyte colony-stimulating factor; GM-CSF, granulocyte-macrophage colony-stimulating factor; IFN, interferon; KC, keratinocyte chemoattractant; MCP-1, monocyte chemotactic protein-1; MIP, macrophage inflammatory protein.

**Table 3 nutrients-14-03872-t003:** The anti-neuroinflammatory effects of GL extracts on central nervous system diseases.

Extracts	Models		Underlying Mechanisms	References
	In Vivo	In Vitro		
Ganoderic acid A (GAA)	D-galactose mice	——	Regulating the imbalance of the Th17/Tregs axis	Zhang Y et al., 2021 [35]
GAA	——	LPS-stimulated BV-2	Activating farnesoid X receptor (FXR)	Jia Y et al., 2021 [37]
GAA	Multiple sclerosis animal	——	Activating farnesoid X receptor (FXR)	Jia Y et al., 2021 [38]
GAA	Post-stroke depression	——	Regulating M1/M2 microglial polarization by activating the ERK/CREB pathway	Zhang L et al., 2021 [39]
Ganoderterpene A	——	LPS-stimulated BV-2	Suppressing the activation of MAPK and TLR-4/NF-κB signaling pathways	Kou RW et al., 2021 [40]
Ganoderma lucidum polysaccharides (GLPs)	——	LPS- and Aβ42-stimulated BV-2 and primary mouse microglia	Modulate microglial phagocytosis and behavioral response	Cai Q et al., 2017 [34]
GLPs	D-galactose rats	——	Regulating inflammation of the brain–liver axis	Zhang Y et al., 2021 [35]
GLPs	Spinal cord ischemia–reperfusion injury	——	Reducing lipid peroxidation, inflammatory cytokine production	Kahveci R et al., 2021 [36]
Ganoderma lucidum triterpenoids (GLTs)	Maternal separation-induced anxiety and depression	——	Reversing up-regulation of pro-inflammatory markers in the periphery and brain, and activating microglia in the prefrontal cortex and hippocampus	Mi X et al., 2022 [7]
The aqueous extract of GL	Kainic acid-induced seizures	——	Decreasing immunoreactivity for GFAP as well as TNF-alpha and IL-1beta in the CA3 region	Aguirre Moreno AC et al., 2022 [42]
GL extracts	——	LPS- and MPP(+)-treated MES23.5 cell	Preventing the production of microglia-derived proinflammatory and cytotoxic factors	Ding H et al., 2010 [11]
GL extracts	——	LPS and MPP(+)-treated co-cultures of microglia and MES 23.5 Cells	Preventing the production of proinflammatory factors	Zhang R et al., 2011 [10]
Deacetyl ganoderic acid F	LPS-stimulated Zebrafish and mice	LPS-stimulated BV-2	Suppression of NO production and pro-inflammatory cytokine secretion, modulation of the NF-κB pathway	Sheng F et al., 2019 [41]

## Data Availability

Not applicable.

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
