# Peer review of "Ganoderma lucidum Modulates Inflammatory Responses following 1-Methyl-4-Phenyl-1,2,3,6-Tetrahydropyridine (MPTP) Administration in Mice"

_nutrients, 2022, doi:10.3390/nu14183872_

Round 1

Reviewer 1 Report

Title: Ganoderma Lucidum should be italic and species is low cap. Ganoderma lucidum must be checked for the whole manuscript

Abstract: In vitro should be italic, please check others too

Introduction: please add one paragraph explaining important bioactive compounds from GLE which responsible for inflammation or antioxidant. Eg. https://doi.org/10.1016/j.carbpol.2021.118231 ,  https://link.springer.com/article/10.1007/s10068-017-0021-6 , https://doi.org/10.1039/D1FO00355K

Safety or non-toxic of GLE. Eg. https://doi.org/10.1016/j.aqrep.2020.100322 , https://doi.org/10.3390/ijms22041675

Methodology: GLE preparation is unclear, please elaborate, picture of GLE is crucial

Results" Figure 1,2 are too small, please emphasize (add arrow too for intended description), refer for clear cells pic https://doi.org/10.4014/jmb.1510.10018

Discussion:

Provide a comparison table of available anti-neuroinflammatory of GLE in vivo, beside Parkinsonian pathogenesis. Highlight your main findings

Provide Graphical abstracts

Mice, GLE, and GLE should be included in a Figure to attract readership

Reviewer 2 Report

Here are the specific suggestions:

1.      What are the other ingredients in the GLE besides polysaccharide and ergosterol? Are those components affect the neuroinflammation of PD and release of cytokines?

2.      Line 83, ‘After one week’ acclimatization’ revised to ‘Acclimatization for one week’.

3.      Line 84, ‘MPTP (sigma, 20 mg/kg, four times at 2 h interval)’, are there only four injections or continued for the entire experimental period? Make the describe more clearly, please.

4.      Line 83-87, The description of experimental design is not detailed enough, please rewrite section 2.3.

5.      Line 111 and 115, There are same section number in these two lines, please revised.

6.      Line 145-147, This sentence is not meet the English writing standards, please revised.

7.      Line 161, Is the model group means MPTP treated group? If yes, please add the describe in materials and method (section 2.3).

8.      Table 1 and table 2, Please clarify the meaning of sign (* and #) in the table, such as under the figures.

9.      Is there significant difference existed in cytokines between control group and the MPTP + GLE group?

Round 2

Reviewer 1 Report

Comments have been corrected. however the species in the title should be lucidum instead of Lucidum

Author Response

We thanks the reviewer again for his/her pertinent comments sincerely. The title and all other places have been corrected. 

Reviewer 2 Report

The author has made a good revised for the unclear and the questions in the manuscript, but there still some questions. Here are the specific suggestions:

1.     “80% ethanol twice, extraction time of 2 h, extraction temperature of 100℃ and then extracted with 50% ethanol twice. The mixture was then extracted using water twice. After centrifugation, the supernatant was centrifuged and lyophilized to dryness.” Is the extraction temperature for 80% ethanol and 50% ethanol both 100℃? And which method was used in these two steps for keep the temperature? After extraction with 80% ethanol once, which part used for the next step? Supernatant or residue? The composition of the mixture used for water extract is not clear, I am not sure if it made of the extraction supernatant in step one and two.

2.     Line 288-296: those sentences were discussed about the Ganoderma lucidum polysaccharides, but the GLE was used in the treatment, it can’t prove the function of Ganoderma lucidum polysaccharides since it is a mixture. Please revised this part.

3.     The data in Table 3 should be write in the same place where the extraction (Ganoderic acid A, Ganoderma lucidum polysaccharide) discussed in this paper. The Ganoderterpene A, Ganoderma lucidum triterpenoids, the aqueous extract of GL, and deacetyl ganoderic acid F could be put into the last paragraph line 344-349.
